# OpenReview forum: "AI Debate Aids Assessment of Controversial Claims"
_NeurIPS.cc/2025/Conference — NeurIPS 2025 poster_

### Official Review · Reviewer_W11V · 2025-06-16

**Clarity:** 3
**Significance:** 2
**Originality:** 2
**Rating:** 3
**Confidence:** 4

**Summary:**

In the context of approaches to scaling oversight of large AI models such as “frontier models performing tasks beyond human capabilities,” the paper aims to gauge approaches that can help ensure that such oversight is reliable, and that human or AI agent judges involved in conducting oversight tasks are not swayed by the models themselves toward incorrect judgements. Through two studies, the research attempts to measure and compare two frameworks for scalable oversight, Debate and Consultancy; One study used human judges and the other used automated judges.

**Questions:**

-Have the authors considered adjusting their conclusion to temper their assessment that the work has implications for high-stakes decisions? High stakes decisions could go far beyond public health related information.
-Have the authors considered providing more detail on why they considered a judge’s background and beliefs? It would be helpful here to more directly state the reasoning behind this, even if it seems obvious.

**Ethical Concerns:**

["NO or VERY MINOR ethics concerns only"]

**Final Justification:**

Following discussion with the authors and analysis of other reviews, particularly several important limitations/weaknesses of the work raised by reviewer bvi9, I maintain my original ratings for the paper and suggest the authors consider the non-trivial societal impacts of automating - in a 'scalable' manner - decisions regarding accuracy of information related to highly-contested and belief-oriented concepts.

**Limitations:**

The authors seem to recognize that their criteria for selecting datasets to conduct the research creates important limitations, noting that "Few datasets exist satisfying all of these criteria." The choice of COVID-19 as the subject area for study presents problems, in part because the topic still remains unsettled in many ways. There do exist fact-checking datasets that include data from earlier years that could be more suitable.

Ultimately, the societal impacts of automating decisions regarding accuracy of information related to highly-contested and belief-oriented concepts is not only a challenge, it could have negative impacts on people that are not considered by the authors. While the study is a small way of understanding the caveats, the authors could benefit from further analysis of the very goal of this work.

**Paper Formatting Concerns:**

No formatting problems noticed by this reviewer.

**Quality:**

3

**Strengths And Weaknesses:**

Strengths
The authors are clear in their goals to study the efficacy of emerging frameworks for scaling oversight of large AI models in the context of problems related to the more widespread technique of reinforcement learning with human feedback. Their work supplements prior work in the area of gauging the Debate protocol introduced in 2018, particularly in comparison to the Consultancy framework. The authors cite related prior work and offer a helpful section detailing how it relates to their research. Their work also attempts to consider and address problems that may arise from manipulation of human and automated judges.

Weaknesses
The paper draws on prior work  in important and potentially problematic ways; some key prior work either has not been peer-reviewed or is peer-reviewed but itself based on non-peer reviewed work. The area of scalable oversight does not appear to be researched robustly yet and may benefit from genuine peer review of “foundational” papers before researchers continue to attempt to advance it.

Another weakness: The authors conclude that their work could have greater implications than the research, which offers a relatively small sample size, may warrant. Despite the fact that their research studied application of the Debate and comparison frameworks only in the context of COVID-19 related information, and in a relatively limited capacity, they conclude that the Debate framework could be “a practical and scalable mechanism for oversight in high-stakes, belief-sensitive domains.”

Also, the authors also cite a paper used in relation to their dataset discovery [50], which appears to have been withdrawn by its authors (https://openreview.net/forum?id=Jztt1nrjAM). Although an updated version of the paper exists (https://arxiv.org/pdf/2411.05060), they'll need to clarify and possibly update the citation.

---

> ### Author Rebuttal · Authors · 2025-07-30
>
> We thank reviewer W11V for their careful review and appreciation of our **clear goals**, **contributions to prior Debate protocol** research, helpful **related work section**, and efforts to **address manipulation problems** in judges. We address all concerns about foundational work citations, generalization claims, dataset choice, and societal impacts below.
>
> ---
>
> ## Q: Limited generalization beyond COVID-19
>
> To address reviewer concerns about generalization beyond COVID-19, we conducted an extensive additional study with human judges on climate change claims.
>
> ### **New Climate Change Study**
>
> To demonstrate generalization, we recruited **146 participants** (73 per belief group) from an initial pool of 1,000 based on their climate change beliefs (human-caused vs. natural cycles), yielding **421 debate sessions** and **424 consultancy sessions** after quality filtering. Following the reviewer's suggestion to use other fact-checking datasets, we selected claims from the **Climate-Fever dataset** [1,2], which contains settled claims backed by scientific evidence, using GPT-4o screening followed by human curation to ensure they satisfy our scalable oversight criteria (details in Section 2.1.1 and Appendix B.1).
>
> **In this study, we use an even stronger consultancy baseline** by allowing consultants to choose which position to argue rather than random 50/50 assignments. Now consultants argue for the correct answer ~92.54% of the time (giving consultants significant advantage), creating a much more challenging comparison that better reflects real-world scenarios where AI assistants like ChatGPT naturally provide their best assessment rather than randomly arguing for correct or incorrect positions.
>
> ### **Results**
>
> **Table: Human judge accuracy before and after debate versus consultancy interventions for climate claims**
>
> *Belief Groups:*
> - **Belief Group 1**: Climate change is primarily caused by natural cycles and variations
> - **Belief Group 2**: Climate change is primarily caused by human activities
>
> | Group | Debate Before → After | Consultancy Before → After | Debate Advantage |
> |-------|----------------------|----------------------------|------------------|
> | **Overall** | 64.5% → **72.1%** (+7.6pp) | 65.8% → **68.3%** (+2.5pp) | **+3.8pp** |
> | **Belief Group 1** (Natural) | 51.2% → **63.9%** (+12.7pp) | 53.0% → **59.2%** (+6.2pp) | **+4.7pp** |
> | **Belief Group 2** (Human-caused) | 76.8% → **80.2%** (+3.4pp) | 74.2% → **75.6%** (+1.4pp) | **+6.0pp** |
>
> Debate achieves **72.1% accuracy** compared to stronger consultancy's **68.3% accuracy**—a **3.8% improvement**. This shows debate's advantages generalize beyond COVID-19 to other contentious scientific domains even with a substantially stronger consultancy baseline.
>
> Additionally, we conducted LLM judge studies on climate claims (detailed in Appendix D.2) which show similar patterns—debate provides better performance than consultancy across both COVID-19 and climate domains, with debate particularly benefiting weaker judges like Qwen-2.5-8B.
>
> We will update Figure 1 to include these climate results and revise Section 2.1.2 to clearly describe our stronger consultancy protocol, directly addressing the reviewer's concerns about generalization beyond COVID-19.
>
> ---
>
> ## Q: Reasoning for considering judge backgrounds
>
> Thank you for highlighting this. We should indeed be more explicit about our rationale for considering judge backgrounds and beliefs.
>
> We recruited judges with different prior beliefs for three key reasons:
>
> **First**, the cognitive science literature documents how confirmation bias and motivated reasoning affect judgment [8,9]. In factual disputes about COVID-19, someone who believes vaccines are safe will evaluate evidence differently than a skeptic—not just in conclusions but in which sources they trust and how they weigh conflicting evidence.
>
> **Second**, real-world AI oversight involves diverse judges with existing worldviews, not neutral evaluators. Understanding how debate performs across this diversity is crucial for practical deployment.
>
> **Third**, this directly addresses the scalable oversight challenge. As noted in our introduction, humans exhibit cognitive biases that lead them to accept arguments aligning with their mental models. We empirically test whether debate helps people assess controversial belief-sensitive claims despite these biases.
>
> We will add a clear statement in Section 2.1.3: *"We recruited judges with different prior beliefs because real-world deployment must account for human belief diversity [10]. Testing debate's effectiveness across belief systems—rather than assuming idealized rational judges—is essential for building oversight mechanisms that work with actual humans."*
>
> ---
>
> ## Q: Tempering claims about generalization to high-stakes domains
>
> Thank you for this important critique. To address this concern, we have conducted additional experiments on climate change claims, expanding our evidence beyond COVID-19. While our results now demonstrate debate's effectiveness across two controversial scientific domains (COVID-19 and climate change), we will acknowledge in the paper that each high-stakes domain has unique characteristics that require careful empirical validation.
>
> Following the reviewer's suggestion, we will revise our conclusion (lines 375-376) from "a practical and scalable mechanism for oversight in high-stakes, belief-sensitive domains" to "*a promising approach for oversight in belief-sensitive factual assessment tasks.*" We will also add explicit caveats about the limitations of generalizing from public health and climate claims to other high-stakes decisions, leaving detailed exploration of additional domains as future work.
>
> ---
>
> ## Q: Foundational work and peer review concerns
>
> We understand the reviewer's concern about the peer review status of foundational work in scalable oversight. While it's true that some early influential papers like Irving et al. [3], Bowman et al. [4], and Michael et al. [5] were initially published as preprints, we respectfully note that subsequent work building on these foundations has been rigorously peer-reviewed at top-tier venues. For example:
>
> - Khan et al. [6] won the **Best Paper Award at ICML 2024**
> - Kenton et al. [7] was published at **NeurIPS 2024**
>
> The field is growing with an increasing number of peer-reviewed papers validating and extending these early ideas. Our work contributes to this growing body of peer-reviewed research by being the first to examine scalable oversight in belief-sensitive domains where judges hold strong prior biases—a critical gap that previous work has not addressed.
>
> ---
>
> ## Q: Citation issue with withdrawn paper [50]
>
> The reviewer is correct that reference [50] has an updated version. The KDD '25 accepted version became available on June 18, 2025, which was after the NeurIPS submission deadline (May 15, 2025). We will update this citation in our camera-ready version.
>
> ---
>
> ## Q: Societal impacts concern
>
> We appreciate the reviewer's important concern about the societal impacts of our work.
>
> From the outset, we deliberately limited our experiments to scientific claims where there is clear, verifiable ground truth, rather than political or values-laden topics. Likewise, our debate protocol was designed to surface evidence and counter-evidence in service of truthfulness, not to maximize persuasion or engagement. Nonetheless, we fully agree that any system capable of influencing beliefs must be examined through an ethical lens.
>
> To make this scrutiny explicit, we will add a dedicated **"Broader Impacts and Ethical Considerations"** section discussing:
> - Potential misuse if debate systems are deployed with misaligned incentives
> - Risks that our findings about persona-based persuasion could enable targeted manipulation
>
> Finally, we will outline concrete safeguards to ensure debate remains an instrument for accurate judgment rather than exploitation. These include:
> - Auditing and transparency requirements for source materials
> - Routine human-in-the-loop checks
> - Clear guidelines on when and where debate-based oversight is appropriate
>
> By articulating these principles and boundaries, we aim to preserve human agency in high-stakes decisions and help guide the responsible deployment of AI debate systems.
>
> ---
>
> We hope our comprehensive responses—the new climate study with strengthened baseline, clarification about foundational work and updated citation, explicit rationale for judge backgrounds, tempered claims about high-stakes implications, and expanded discussion of societal impacts with safeguards—fully address the reviewer's concerns. If the reviewer finds these improvements satisfactory, we would be grateful for their consideration in revisiting the score.
>
> ### References
>
> 1. Diggelmann et al., "Climate-fever: A dataset for verification of real-world climate claims", arXiv 2020.
> 2. Gabriel et al., "Misinfo reaction frames: Reasoning about readers' reactions to news headlines", ACL 2022.
> 3. Irving et al., "AI safety via debate", arXiv 2018.
> 4. Bowman et al., "Measuring Progress on Scalable Oversight for Large Language Models", arXiv 2022.
> 5. Michael et al., "Debate Helps Supervise Unreliable Experts", arXiv 2023.
> 6. Khan et al., "Debating with More Persuasive LLMs Leads to More Truthful Answers", ICML 2024.
> 7. Kenton et al., "On scalable oversight with weak LLMs judging strong LLMs", NeurIPS 2024.
> 8. Kahneman, "Thinking, fast and slow", 2011.
> 9. Kunda, "The case for motivated reasoning", Psychological Bulletin 1990.
> 10. Irving and Askell, "AI safety needs social scientists", Distill 2019.

---

> > ### Comment · Reviewer_W11V · 2025-08-01
> >
> > Thanks to the authors for considering the review comments.
> >
> > It is helpful that they conducted the additional climate change study. It is also helpful that the authors will alter their conclusion to recognize the limits of their work.
> >
> > As for the concerns regarding the foundational research this paper attempts to build on, although some key prior scalable oversight work has been peer-reviewed, the foundations of that prior work are themselves not. Much of the prior work referenced in Khan et al, for example, is not peer-reviewed. This makes for a rickety foundation upon which to stand up new work.

---

> > > ### Author Response · Authors · 2025-08-04
> > >
> > > Thank you for acknowledging our additional climate change experiments and revised conclusions.
> > >
> > > We want to clarify that our empirical findings are entirely self-contained. Our extensive experiments across multiple domains provide rigorous validation of the debate protocol—**no conclusions depend on accepting claims from non-peer-reviewed work**.
> > >
> > > While we cite foundational scalable oversight papers for context, **we cannot ignore existing work in this nascent field simply because it lacks peer review**. Our empirical results require no theoretical assumptions from these works and can be evaluated purely on their own merits.
> > >
> > > We hope this addresses your concern.

---

> > > > ### Comment · Reviewer_W11V · 2025-08-08
> > > >
> > > > Thanks for clarifying that your conclusions do not rely on the non-peer reviewed work cited in the paper. It would be helpful to consider why seemingly foundational prior work in the field lacks peer review.
> > > >
> > > > It is also notable that the additional study approach, as you state, "giving consultants significant advantage" creates "a much more challenging comparison that better reflects real-world scenarios where AI assistants like ChatGPT naturally provide their best assessment rather than randomly arguing for correct or incorrect positions."
> > > >
> > > > An additional question I would consider is why, if this does indeed "better" reflect real-world scenarios, it was not the approach used in the study highlighted in the original version of the paper. Was the goal to find the most favorable result for the Debate approach or to reflect real world scenarios?

---

### Official Review · Reviewer_bvi9 · 2025-06-30

**Clarity:** 3
**Significance:** 2
**Originality:** 2
**Rating:** 3
**Confidence:** 5

**Summary:**

While scalable oversight methods have typically been conceptualized in the context of training and evaluation of AI models for correctness/truthfulness, the same mechanisms could be considered in the context of improving the information landscape in general. This paper touches on this goal, specifically for evaluating the correctness of misinformation/debunking misinformation. They focus on COVID-19 claims, and ran studies with human judges (sampling from both those who believed the mainstream claim and those who were skeptical of it) as well as LLM judges (prompted with a "persona" priming it to have biases matching the human sample) comparing Debate against Consultancy for effectiveness at improving the accuracy of judges' beliefs (i.e. to match the mainstream view).

They find that:
- Debate is better than Consultancy at improving judge accuracy.
- The effect is especially stronger for Judges who already hold the correct belief (i.e. the impact in strengthening mainstream believers' convictions is more than the impact in making skeptics switch beliefs)

**Questions:**

1) What is the explanation for focusing on COVID-19 claims?
2) To what extent do you think the results are influenced by the relative _prior convictions_ in each belief? --- it seems very likely that people who hold non-mainstream beliefs are less sure of them.
3) I am interested to know how the educational background of the believers affects their susceptibility to persuasion --- this is also a confounding factor, but perhaps one you could adjust for from your existing data.
4) I am also interested by your finding that the belief update is greater for those who already hold the mainstream belief --- this would indicate that debate would widen the belief gap. Do you think this might be a factor affecting political polarization in the real world?

**Ethical Concerns:**

["NO or VERY MINOR ethics concerns only"]

**Final Justification:**

The minor comments/weaknesses I listed have been addressed:
- Background literature not referenced -- these have now been cited.
- Consultancy baseline is weak -- the baseline has been replaced in the new experiment by a much stronger "Open Consultancy" baseline, where the participant may choose to argue for either the true or false side (and in practice tends to argue for the true side), and Debate is still observed to outperform it
- Mediating factors (prior convictions, educational background) -- these questions have been satisfactorily addressed
- Polarization -- this question has been satisfactorily answered

However, the central issue with the paper remains. To reiterate: in the chosen domain (COVID-19 claims), the true claims correspond exactly to the *mainstream* claims -- this makes "being mainstream" a confounding factor mediating truth and persuasion, i.e. the results can be explained simply by the mainstream view having more arguments backing it in the literature.

The authors did a subsequent study in a different domain (climate change claims); however, this confounding remains true for climate change claims.

The authors have argued that giving both debaters comparable source pools controls for this confounding factor -- however, this strikes me as false, because the debaters' responses will not be restricted to the information contained in the sources provided in-context. For example: even if two human debaters were provided comparable-quality source material to refer to in their debate, they would still have the entire sum of their background knowledge to tap on.

I see two possible ways a better experiment could be done:
1) Continue with a similar experiment design, but choose a domain where the debaters do not have any background information: for example, misinformation on the contents of some recent report or bill.
2) The suggestion I made in my original review:

> One robust way to select a better sample: look at controversial claims being traded on prediction markets or metaculus, e.g. claims about ongoing wars, causes for accidents, or back when the LK-99 superconductor story was in the news, or COVID-19 questions where they were yet unresolved, or Biden's health issues --- all of these were controversial questions whose ground truth would eventually be (but were not yet) resolved. Do the experiment with these claims now, then add ground truth labels once the markets resolve. This way you can also evaluate the confounding effect of "mainstream-ness" separately.

**Limitations:**

I am not familiar enough with ethical stuff to say for certain, but the collection of demographic info from the human participants to create "LLM personas" of them might be a cause for concern. The authors say:

> This study was performed under approval from the appropriate institutional ethics review board. Full IRB documentation will be made available upon request.

It might be worth requesting this.

**Quality:**

1

**Strengths And Weaknesses:**

*Strengths*

1) The application of Debate to evaluating information in "the wild" is novel to my knowledge; while existing works focus on evaluating AI models.
2) The study accounts for the problem for the judges having preconceived biases (which is a problem unique to this domain; judges would not have preconceived biases over a question-answering task for example) and evaluate the cases of the judge being preconceived toward each side separately.
3) They also account for this in their experiment with LLM judges, using persona-based LLMs (based on the demographics of the people in their data who hold each belief) to simulate biased judges.

*Weaknesses*

1) The most serious weakness is in the sample of "misinformation claims": for example, it is possible that AI debate is good at convincing people against /non-mainstream/ viewpoints rather than /false/ ones (one obvious hypothesis for why this might be: mainstream claims have a larger body of literature backing them than non-mainstream claims), and the chosen sample (COVID-19 claims etc.) simply leads to confounding between these features.

   One robust way to select a better sample: look at controversial claims being traded on prediction markets or metaculus, e.g. claims about ongoing wars, causes for accidents, or back when the LK-99 superconductor story was in the news, or COVID-19 questions where they were yet unresolved, or Biden's health issues --- all of these were controversial questions whose ground truth would eventually be (but were not yet) resolved. Do the experiment with these claims now, then add ground truth labels once the markets resolve. This way you can also evaluate the confounding effect of "mainstream-ness" separately.

   The paper consider the problem of the judges having preconceived biases, but not of biases in the dataset.

2) In fact the paper seems to have /only/ looked at COVID-19 claims as their misinformation sample. This severely limits generalization!

3) Not enough engagement with existing literature applying scalable oversight to misinformation/etc. There is some previous work in this area, e.g. https://www.science.org/doi/10.1126/science.adq1814 which has not been cited. (Incidentally this paper also reinforces the point in 1).

4) The consultancy baseline is quite weak --- qualitatively the result only demonstrates that "hearing both sides is better than hearing the wrong side 50\% of the time", which is quite an expected result. This weakness (even in previous debate experiments) is pointed out in https://arxiv.org/pdf/2504.03731 who propose a more robust benchmark.

---

> ### Author Rebuttal · Authors · 2025-07-30
>
> We thank reviewer bvi9 for their thorough review. We appreciate that the reviewer recognizes **the novelty of applying Debate to real-world information evaluation**, our **careful consideration of judge biases**, and our **innovative use of persona-based LLMs**. We address all the reviewer's concerns below.
>
> ---
>
> ## Q: Explanation for COVID-19 focus
>
> We selected COVID-19 claims because they uniquely satisfy our requirements: (1) people hold strong, divergent prior beliefs (mainstream vs. skeptical) that significantly affect judgment, and (2) the selected claims meet established scalable oversight criteria as detailed in Section 2.1.1 and Appendix B. These claims also have direct societal impact - as noted in lines 2-4, misinformation on public health topics can cause real harm, making it critical to study how AI debate can help people evaluate facts despite their biases. We will clarify this in Section 2.1.1.
>
> ---
>
> ## Q: Sample selection, mainstream vs. false confound, and dataset bias
>
> Thank you for this thoughtful critique.
>
> We carefully selected claims through multiple rounds of GPT-4o screening and human curation to ensure claims have ground truth backed by scientific evidence. In our study, "mainstream" views align with scientific consensus (i.e., factually correct positions), while "skeptical" views diverge from it. Our goal is to examine whether debate helps people reach accurate judgments on controversial claims despite their prior beliefs. The claims satisfy scalable oversight criteria—controversial yet with verifiable ground truth from scientific literature—allowing us to evaluate protocol effectiveness.
>
> We appreciate the reviewer's suggestion about using prediction market claims where ground truth emerges later. While such longitudinal studies would be valuable (requiring long waiting periods), we conducted an immediate additional study on climate change claims to demonstrate debate's effectiveness across different domains. We will include prediction markets in our future work section, acknowledging that such studies with initially unresolved claims could provide additional insights into separating mainstream bias from truth-seeking mechanisms.
>
> ---
>
> ## Q: Limited generalization beyond COVID-19 and weak consultancy baseline
>
> To address reviewer concerns about generalization beyond COVID-19 and baseline strength, we conducted an extensive additional study with human judges on climate change claims.
>
> ### **New Climate Change Study with Stronger Baseline**
>
> To demonstrate generalization, we recruited **146 participants** (73 per belief group) from an initial pool of 1,000 based on their climate change beliefs (human-caused vs. natural cycles), yielding **421 debate sessions** and **424 consultancy sessions** after quality filtering. We selected claims from the Climate-Fever dataset [1] using GPT-4o screening followed by human curation to ensure they satisfy our scalable oversight criteria (details in Section 2.1.1 and Appendix B.1).
>
> **Critically, we strengthened the consultancy baseline** by allowing consultants to choose which position to argue rather than random 50/50 assignments. This addresses the reviewer's concern that our original baseline was artificially weak. Now consultants argue for the correct answer ~92.54% of the time (giving consultants significant advantage), creating a much more challenging comparison that better reflects real-world scenarios where AI assistants like ChatGPT naturally provide their best assessment rather than randomly arguing for correct or incorrect positions.
>
> ### **Results**
>
> **Table: Human judge accuracy before and after debate versus consultancy interventions for climate claims**
>
> *Belief Groups:*
> - **Belief Group 1**: Climate change is primarily caused by natural cycles and variations
> - **Belief Group 2**: Climate change is primarily caused by human activities
>
> | Group | Debate Before → After | Consultancy Before → After | Debate Advantage |
> |-------|----------------------|----------------------------|------------------|
> | **Overall** | 64.5% → **72.1%** (+7.6pp) | 65.8% → **68.3%** (+2.5pp) | **+3.8pp** |
> | **Belief Group 1** (Natural) | 51.2% → **63.9%** (+12.7pp) | 53.0% → **59.2%** (+6.2pp) | **+4.7pp** |
> | **Belief Group 2** (Human-caused) | 76.8% → **80.2%** (+3.4pp) | 74.2% → **75.6%** (+1.4pp) | **+6.0pp** |
>
> Debate achieves **72.1% accuracy** compared to stronger consultancy's **68.3% accuracy**—a **3.8% improvement**. This shows debate's advantages generalize beyond COVID-19 to other contentious scientific domains even with a substantially stronger consultancy baseline.
>
> Additionally, we conducted LLM judge studies on climate claims (detailed in Appendix D.2) which show similar patterns—debate provides better performance than consultancy across both COVID-19 and climate domains, with debate particularly benefiting weaker judges like Qwen-2.5-8B.
>
> We will update Figure 1 to include these climate results and revise Section 2.1.2 to describe our stronger consultancy protocol, addressing both concerns.
>
> ---
>
> ## Q: Missing literature
>
> Thank you for highlighting Costello et al. [2] - we cite it in line 233 (reference 14) and use their persuasion taxonomy. We agree we should better position our work at the intersection of scalable oversight and misinformation. While previous scalable oversight studies focus on reading comprehension tasks [3, 4], and misinformation work primarily examines single-advisor consultancy setups [2, 5], our debate approach represents a novel approach. As per your suggestion, we will expand our discussion in Related Work (Section 4, "Scalable oversight" paragraph) and Introduction (Section 1) to highlight how debate's adversarial structure might address some limitations of consultancy-based approaches for controversial claims.
>
> ---
>
> ## Q: Educational background and susceptibility to persuasion
>
> Thank you for this insightful suggestion. In our main paper (lines 207-228), our mixed-effects models already control for LLM experience, which correlates with educational background. However, to directly address reviewers' comments about education, we conducted additional analysis examining how educational background affects susceptibility to persuasion (i.e., likelihood of changing one's initial belief) across our 896 sessions with 68 participants on COVID-19 claims.
>
> ### Our findings:
>
> **Education significantly moderates debate's effectiveness:** Higher education enhances debate's protective effect against belief manipulation. PhD holders showed the strongest resistance—only 15.9% changed their beliefs in debate versus 36.2% in consultancy (a 20.3 percentage point protective effect).
>
> **Debate reduces education-based disparities:** In consultancy, belief change rates vary widely across education levels (from 31.4% for undergraduates to 58.3% for secondary education). In debate, rates are more consistent across all education levels (15.9% to 29.7%), suggesting debate helps equalize outcomes regardless of educational background.
>
> This analysis reveals that while participants with advanced degrees benefit most from debate, the protocol improves judgment across all education levels. We will include this educational breakdown in our revised manuscript.
>
> ---
>
> ## Q: Influence of prior conviction strength between belief groups
>
> We analyzed initial confidence levels and found both groups hold similarly strong convictions: mainstream believers averaged **70.1% confidence** vs skeptical believers at **68.1%**—essentially very small difference. This contradicts the hypothesis that non-mainstream believers are less sure of their beliefs. Figure 3 details how initial confidence levels (Low/Moderate/Strong) impact protocol effectiveness, showing debate benefits apply regardless of belief type.
>
> ---
>
> ## Q: Debate's differential impact and implications for polarization
>
> The reviewer is correct that debate strengthens mainstream believers' accuracy more than it converts skeptics. However, it's crucial to note that **both groups move toward truth**—mainstream believers improve from 64.2% to 71.1% accuracy, while skeptics improve from 62.0% to 69.0%. The "gap" reflects differential movement toward accuracy, not increased polarization where groups move apart.
>
> Our climate study shows similar patterns: both belief groups improve accuracy with debate, though mainstream believers benefit more. This suggests the finding generalizes beyond COVID-19.
>
> Regarding real-world political polarization, the key distinction is **incentive structure**. Our debate protocol explicitly optimizes for truth-seeking with verifiable ground truth, which is why we chose scientific over political claims—the latter often lack verifiable ground truth and raise ethical concerns about belief manipulation. We'll add a "Broader Impacts" section discussing how debate systems need truth-seeking incentives to avoid polarization.
>
> ---
>
> ## Q: IRB and ethical considerations
>
> Our IRB-approved study collects only broad demographic categories (age ranges, location type, education level, income brackets, political/religious affiliations) - **No personally identifiable information is collected.** We ensure complete anonymity through aggregate profiles and obtain informed consent. We will add these safeguards to Section 2.1.3 and are happy to share full IRB documentation.
>
> ---
>
> We hope these comprehensive responses fully address the reviewer's concerns and would be grateful for their consideration in revisiting the score.
>
> ### References
>
> 1. Diggelmann et al., "Climate-fever: A dataset for verification of real-world climate claims", arXiv 2020.
> 2. Costello et al., "Durably reducing conspiracy beliefs through dialogues with ai", Science 2024.
> 3. Michael et al., "Debate Helps Supervise Unreliable Experts", arXiv 2023.
> 4. Khan et al., "Debating with More Persuasive LLMs Leads to More Truthful Answers", ICML 2024.
> 5. Costello et al., "Just the facts: How dialogues with ai reduce conspiracy beliefs", 2025.

---

> > ### Author Response · Authors · 2025-08-04
> >
> > Dear Reviewer bvi9,
> >
> > We sincerely hope our detailed response addressing your concerns about generalization, baseline strength, additional analyses, and all other points raised has been helpful. We would be grateful for any thoughts on our clarifications and new results.
> >
> > Thank you for your consideration.
> >
> > Best regards,
> >
> > Authors of Submission 20482

---

> > > ### Author Response · Authors · 2025-08-08
> > >
> > > Dear Reviewer bvi9,
> > >
> > > As the discussion deadline approaches tomorrow (August 8), we wanted to briefly check if you had any questions about our comprehensive response, including our additional full-scale experiment on climate change claims with a stronger consultancy baseline, our clarifications addressing your concerns about mainstream versus false claims, our analyses on educational background effects and prior conviction strength between belief groups, and our responses to your other concerns regarding missing literature and ethical considerations.
> > >
> > > If our clarifications are satisfactory, we would be grateful for your consideration in revisiting the score.
> > >
> > > Thank you for your valuable feedback.
> > >
> > > Best regards,
> > >
> > > Authors of Submission 20482

---

> > ### Comment · Reviewer_bvi9 · 2025-08-09
> >
> > Thank you for your comprehensive response -- most of my minor comments (consultancy baseline, missing literature, questions over results being moderated by factors like education) have been addressed.
> >
> > However, the central flaw in the paper remains: to reiterate, you have chosen a domain (COVID-19) where the correct view is exactly the "mainstream view", thus the results can be explained merely by the correct view having more arguments backing it in the literature. The additional experiment you did (with climate change claims) only doubles down on this.

---

> ### Author Response · Authors · 2025-08-09
>
> Dear Reviewer bvi9,
>
> We are glad that our response helped address your other concerns. Regarding your main concern that "correct view having more arguments backing it in the literature," **we want to clarify that this is not the case in our experimental design.**
>
> As stated in lines 101-102 of our paper: "*Both protocols provide AI systems with relevant reference sources for evidence-based arguments that use proper citations.*" Importantly, we provide **equal numbers of reference sources** for both correct and incorrect positions to ensure fairness.
>
> As detailed in Appendix B.3, we carefully curated source pools for both sides:
> - For correct claims: We provided factual sources
> - For incorrect claims: We provided sources that appear reliable but support incorrect positions, collected through targeted searches across various domains
>
> The key point is that misinformation sources were specifically selected to be **plausible and convincing**, not merely obvious misinformation. We ensured both sides had substantial, citable material of comparable quality and quantity. This design choice eliminates the advantage that correct positions might have from simply having "more arguments in the literature."
>
> To make this clearer, we will expand Section 2.1.2 to explicitly detail:
> 1. Equal number of reference sources provided to both positions
> 2. The careful curation process ensuring misinformation sources appear credible
> 3. How this design controls for any potential literature advantage
>
> This experimental control ensures that debate's success stems from its adversarial mechanism for truth-seeking, not from an imbalance in available evidence.
>
> Thank you for highlighting this important clarification need. We hope this clarification addresses your main concern about the potential confound, **we would be grateful if you would consider revising your score, which would mean a great deal to us.**
>
> Best regards,
>
> Authors of Submission 20482

---

> > ### Comment · Reviewer_bvi9 · 2025-08-09
> >
> > The problem cannot be mitigated by simply providing the AIs with an equal amount of references from both sides, as the AI's responses will not be restricted to the information contained in the sources provided in-context. For example: even if two human debaters were provided comparable-quality source material to refer to in their debate, they would still have the entire sum of their background knowledge to tap on.
> >
> > If you want to control for information, you must choose a domain where the debaters do not have any background information: for example, misinformation on the contents of some report or bill.

---

> > > ### Author Response · Authors · 2025-08-09
> > >
> > > Dear Reviewer bvi9,
> > >
> > > Thank you for your feedback. We would like to clarify the core premise of our study, as there appears to be a fundamental misunderstanding about our research objectives.
> > >
> > > Our work specifically investigates **how debates can enhance decision-making when participants hold differing belief priors**—a critical challenge for scalable oversight. The fundamental premise of scalable oversight is to develop methods for supervising AI systems that may possess capabilities exceeding those of non-expert humans (Bowman et al., 2022; Khan et al., 2024; Amodei et al., 2016). In real-world deployment, human overseers will inevitably bring their own beliefs, biases, and background knowledge when evaluating AI outputs on contentious topics.
> > >
> > > The approach you suggest—focusing on domains where debaters lack background information, such as unfamiliar reports or bills—would not align with scalable oversight goals. Human judges will have pre-existing beliefs that could lead them to incorrectly accept or reject AI-generated information and AI system will be much more capable than human. **Our research addresses precisely this challenge: can debate help humans with strong prior beliefs correctly evaluate AI outputs despite their biases?**
> > >
> > > The value of our research lies in testing debate as a scalable oversight method for scenarios where human judges have pre-existing beliefs and biases, and differing perspectives can be leveraged constructively in debate settings, rather than examining scenarios where all parties begin from equal information deficits.
> > >
> > > We appreciate your engagement with our work and regret that the timing has not allowed for a more extensive discussion of these methodological considerations. We believe our current approach is well-suited to address the research questions we have posed and contributes meaningfully to the literature on scalable oversight.
> > >
> > > We hope this clarification addresses your concerns about our study design.
> > >
> > > Best regards,
> > >
> > > Authors of Submission 20482
> > >
> > > References:
> > > - Amodei et al., "Concrete Problems in AI Safety", arXiv 2016.
> > > - Bowman et al., "Measuring Progress on Scalable Oversight for Large Language Models", arXiv 2022.
> > > - Khan et al., "Debating with More Persuasive LLMs Leads to More Truthful Answers", ICML 2024.

---

> > > > ### Comment · Reviewer_bvi9 · 2025-08-09
> > > >
> > > > > The approach you suggest—focusing on domains where debaters lack background information, such as unfamiliar reports or bills—would not align with scalable oversight goals.
> > > >
> > > > This is not true -- in fact, both the Khan and Bowman papers you mentioned in the previous line perform their experiments in such domains (comprehension/question-answering tasks on the QuALITY and MMLU datasets).
> > > >
> > > > I understand that you want to focus on domains where the human judges already have preconceived biases -- it is possible to find domains where human judges have preconceived biases without having prior info, e.g. people may be more or less inclined to believe claims about a new bill based on their political ideology.
> > > >
> > > > In any case, the fundamental point remains: your results can be explained by the true claim simply having more backing in the literature due to being the mainstream view. To make the claim that AI debate leads human judges to form more truthful beliefs -- surpassing a basic heuristic like "just trust the mainstream view" -- you need to control for this confounding factor.

---

> ### Author Response · Authors · 2025-08-09
>
> Dear Reviewer bvi9,
>
> Thank you for your response. We believe there may be a misunderstanding about the Khan and Bowman papers.
>
> However, Khan et al. explicitly state: "We give the **debaters full access** to the underlying text while judges have no access to the text." The debaters use both the provided text AND **their training knowledge.**
>
> Similarly, Bowman et al. state: "We expect the model to be able to **use significant domain knowledge from pretraining** that our human participants aren't familiar with."
>
> Both papers test information asymmetry (judges lack access to source material) while AI systems **actively use their background knowledge**—exactly as in our study. Neither paper removes **AI's training knowledge**.
>
> Regarding your concern that "results can be explained by the true claim simply having more backing in the literature," we reiterate that we provide equal numbers of reference sources for both correct and incorrect positions. The sources supporting incorrect claims are carefully curated from credible-appearing outlets to ensure they seem as convincing as mainstream sources. Moreover, Sharma et al. (2024) show LLMs can be sycophantic and persuasive even when optimized for truthfulness, Wen et al. (2024) and Hubinger et al. (2024) demonstrate that LLMs' persuasive capabilities persist through safety training, and Salvi et al. (2024) find LLMs can construct compelling persuasive arguments that outperform humans. These findings suggest that **LLMs can construct strong persuasive arguments supporting incorrect positions.**
>
> We respectfully reiterate that testing on topics where AI lacks knowledge would eliminate the very capability asymmetry that scalable oversight must address.
>
> We hope this clarifies our experimental controls and alignment with foundational scalable oversight research.
>
> Best regards,
>
> Authors of Submission 20482
>
> References:
>
> - Sharma et al., "Towards Understanding Sycophancy in Language Models", ICLR 2024.
> - Wen et al., "Language Models Learn to Mislead Humans via RLHF", arXiv 2024.
> - Hubinger et al., "Sleeper Agents: Training Deceptive LLMs that Persist Through Safety Training", arXiv 2024.
> - Salvi et al., "On the Conversational Persuasiveness of Large Language Models", arXiv 2024.

---

> > ### Comment · Reviewer_bvi9 · 2025-08-09
> >
> > > Neither paper removes AI's training knowledge.
> >
> > Obviously I did not claim that any work "removes AI's training knowledge". I was responding to your claim that "focusing on domains where debaters lack background information, such as unfamiliar reports or bills—would not align with scalable oversight goals".
> >
> > The key point is to pick a domain where background knowledge doesn't lead to one side being systematically easier to argue for than the other. For example in a question-answering task in QuALITY, the AI systems may indeed "use" their background knowledge in any way, but you cannot systematically predict that one side will have more arguments in the training knowledge than the other.
> >
> > > We believe there may be a misunderstanding about the Khan and Bowman papers.
> >
> > Those are some of the most relevant references to your work. And other similar empirical works on scalable oversight (e.g. Kenton, Radhakrishnan) also focus on such domains that you claimed "would not align with scalable oversight goals".

---

> > > ### Author Response · Authors · 2025-08-09
> > >
> > > Dear Reviewer bvi9,
> > >
> > > We respectfully clarify that when we said "focusing on domains where debaters lack background information... would not align with scalable oversight goals," we were referring to **domains where debaters have no background knowledge whatsoever**.
> > >
> > > In QuALITY (Khan et al., Michael et al.), there is information asymmetry between judges and debaters, but **debaters still actively use their background knowledge to understand and argue about the stories**. Similarly, Kenton et al. ensure capability gaps while AI systems leverage their knowledge. **We never claimed these approaches don't satisfy scalable oversight criteria.**
> > >
> > > What we stated is: if debaters lack background information, this would violate scalable oversight principles that require AI systems to leverage superior capabilities (Bowman et al., Khan et al.).
> > >
> > > We appreciate your understanding that we focus on "domains where human judges already have preconceived biases." Previous work chose different domains because they were not investigating pre-existing beliefs. Our choice of COVID-19 and climate change domains was deliberate. As we explained earlier: We selected these claims because they (1) involve strong, divergent prior beliefs that significantly affect judgment, and (2) meet established scalable oversight criteria while having direct societal impact. Misinformation on public health and climate topics can cause real harm, making it critical to study how AI debate can help people evaluate facts despite their biases.
> > >
> > > Best regards,
> > >
> > > Authors of Submission 20482

---

### Official Review · Reviewer_feUp · 2025-07-02

**Clarity:** 2
**Significance:** 3
**Originality:** 3
**Rating:** 5
**Confidence:** 3

**Summary:**

The paper investigates whether adversarial debate between large language models (LLMs) can improve human and AI judgment on controversial factual claims, specifically in the context of COVID-19 misinformation. It compares a debate protocol, where two AI agents argue opposing sides of a claim, to a consultancy setup involving a single AI advisor who can be "consulted". Human judges with either "mainstream" or "skeptical" beliefs are evaluated pre- and post-intervention, and a second study compares the performance of persona-based LLM judges against human judges.

The authors find that debate improves factual accuracy more than consultancy. Persona-based LLM judges outperform humans and default LLMs in supervising factual claims. The authors argue that debate promotes better belief updating and confidence calibration, supporting its use as a scalable oversight mechanism.

**Questions:**

- Abstract: It is hard to parse the abstract accurately in isolation without having read the complete paper. There are 2 studies, one with human judges and one with personalized AI judges. And there are two protocols, debate and consultancy. The relationships between the studies, judge types, and protocols are not clearly laid out, making it hard to follow the overall structure and contributions of the work.
- Introduction: I found it difficult to understand the overall setup of the study just from the text. It would be nice if you can come up with some examples or visualization to make this more clear (move from appendix to main paper).
- I recommend revising the main paper to include more methodological transparency by moving the content from appendix to main paper.

**Ethical Concerns:**

["NO or VERY MINOR ethics concerns only"]

**Final Justification:**

Authors have addressed all concerns raised in the review very clearly. I appreciate them including additional experiments to strengthen the claims made in the paper

**Limitations:**

Since the topic is related to deception and persuasiveness, it is important to discuss the impact and potential misuse of such systems.

**Paper Formatting Concerns:**

Please double check if the formatting from lines 277-294 is supported.

**Quality:**

3

**Strengths And Weaknesses:**

Kindly note that my comments are based solely on the content presented in the main paper. While I appreciate that supplementary material can provide additional context, I believe the main paper should contain the core information necessary to understand the experimental design and interpret the results. In its current form, the paper relies heavily on a lengthy appendix (over 60 pages), which places a significant burden on the reader.

### Strengths

- The paper addresses a interesting and valid issue in AI alignment and safety: how to supervise LLMs when humans are biased or less knowledgeable. Motivation for the study is clear and well presented in the paper.
- The use of adversarial AI debate as a tool to correct cognitive bias in factual reasoning is very intuitive.
- The authors conduct two comprehensive studies: one with human judges and another with persona-based LLM judges, covering both human-in-the-loop and AI-only supervision settings. The sample includes both "mainstream" and "skeptical" belief holders, allowing for better analysis of how AI interaction affects different cognitive baselines.


### Weaknesses
- The baseline (consultancy) is very weak in comparison to the debate method, so the fact that debate performs better than baseline is unsurprising.
- The study is limited to one domain. While this is a high-stakes domain, it is hard to asses whether results generalize to other polarized or technical topics (e.g., climate change)
- Most of the questions that come up while reading the paper is actually answered in the appendix. Can you provide a visualization of the study setup? What kind of claims were used? Where legitimate evidences / sources presented by LLM debaters. etc.
- While this is a promising and empirically grounded paper exploring AI debate for oversight, the strongest contributions are undercut by over-reliance on the appendix, missing qualitative depth, and a somewhat weak baseline comparison.

---

> ### Author Rebuttal · Authors · 2025-07-30
>
> We thank reviewer feUp for their constructive feedback. We appreciate that the reviewer finds our work addresses an **interesting and valid issue in AI alignment and safety**, recognizes our **clear and well-presented motivation**, values our **intuitive use of adversarial AI debate** to correct cognitive bias, and acknowledges our **comprehensive two-study design** covering both human-in-the-loop and AI-only supervision with diverse belief groups.
>
> We address the reviewer's concerns about baseline strength, generalization, appendix reliance, and visualization needs below.
>
> ---
>
> ## Q: Abstract clarity, visualization needs, and reducing appendix reliance
>
> We appreciate the reviewer's thorough feedback about improving clarity and accessibility. We understand that the paper's structure and key details need to be more transparent in the main text.
>
> In the final version of the paper, we will revise the manuscript to address all the specific concerns:
>
> ### **Abstract restructuring**
> Following the reviewer's suggestion, we will restructure the abstract to clearly map out:
> 1. **Study 1** examines how human judges with mainstream vs. skeptical COVID-19/climate change beliefs evaluate claims through debate (two AI agents argue opposing sides) vs. consultancy (single AI advisor)
> 2. **Study 2** tests whether AI judges with/without human personas can effectively supervise these same protocols
>
> This will make the relationships between studies, judge types, and protocols immediately clear as requested.
>
> ### **Introduction and visualization**
> Addressing the specific questions about study setup and claim examples, we have prepared a comprehensive overview figure showing: example claims (e.g., "Homemade cloth masks are less effective than surgical ones"), how evidence is presented through web sources with URL citations, complete participant flow from screening to final judgment, and side-by-side protocol comparisons. Due to NeurIPS rebuttal guidelines, we cannot include it here, but this figure will appear prominently on page 2 of the camera-ready version.
>
> ### **Moving core content to main paper**
> Following the recommendation to reduce appendix reliance, we will relocate essential methodological details including:
> 1. Table of representative claims showing the controversial yet verifiable nature of our task
> 2. How LLM debaters/consultants access and cite legitimate web sources during arguments
> 3. Participant recruitment process and demographics
> 4. Detailed protocol specifications from the appendix that are crucial for understanding our work
>
> The camera-ready version will ensure the **main paper stands alone with all information needed to understand and evaluate our methods and findings**. Thank you for this constructive feedback that will improve our manuscript's clarity and accessibility.
>
> ---
>
> ## Q: Limited generalization beyond COVID-19 and weak consultancy baseline
>
> Thank you for raising these important concerns about generalization beyond COVID-19 and the strength of our consultancy baseline. **Following the reviewer's specific suggestion to test climate change claims**, we conducted an extensive additional study with human judges on climate change claims to directly address both issues.
>
> ### **New Climate Change Study with Stronger Baseline**
>
> To demonstrate generalization, we recruited **146 participants** (73 per belief group) from an initial pool of 1,000 based on their climate change beliefs (human-caused vs. natural cycles), yielding **421 debate sessions** and **424 consultancy sessions** after quality filtering. We selected claims from the Climate-Fever dataset [1,2] using GPT-4o screening followed by human curation to ensure they satisfy our scalable oversight criteria (details in Section 2.1.1 and Appendix B.1).
>
> Critically, we **strengthened the consultancy baseline** by allowing consultants to choose which position to argue rather than random 50/50 assignments. This addresses the reviewer's concern that our original baseline was artificially weak. Now consultants argue for the correct answer ~92.54% of the time (giving consultants significant advantage), creating a much more challenging comparison that better reflects real-world scenarios where AI assistants like ChatGPT naturally provide their best assessment rather than randomly arguing for correct or incorrect positions.
>
> ### **Results**
>
> **Table: Human judge accuracy before and after debate versus consultancy interventions for climate claims**
>
> *Belief Groups:*
> - **Belief Group 1**: Climate change is primarily caused by natural cycles and variations
> - **Belief Group 2**: Climate change is primarily caused by human activities
>
> | Group | Debate Before → After | Consultancy Before → After | Debate Advantage |
> |-------|----------------------|----------------------------|------------------|
> | **Overall** | 64.5% → **72.1%** (+7.6pp) | 65.8% → **68.3%** (+2.5pp) | **+3.8pp** |
> | **Belief Group 1** (Natural) | 51.2% → **63.9%** (+12.7pp) | 53.0% → **59.2%** (+6.2pp) | **+4.7pp** |
> | **Belief Group 2** (Human-caused) | 76.8% → **80.2%** (+3.4pp) | 74.2% → **75.6%** (+1.4pp) | **+6.0pp** |
>
> Debate achieves **72.1% accuracy** compared to stronger consultancy's **68.3% accuracy**—a **3.8% improvement**. This shows debate's advantages generalize beyond COVID-19 to other contentious scientific domains even with a substantially stronger consultancy baseline.
>
> Additionally, we conducted LLM judge studies on climate claims (detailed in Appendix D.2) which show similar patterns—debate provides better performance than consultancy across both COVID-19 and climate domains, with debate particularly benefiting weaker judges like Qwen-2.5-8B.
>
> We will update Figure 1 to include these climate results and revise Section 2.1.2 to clearly describe our stronger consultancy protocol, directly addressing the reviewer's concerns about both generalization and baseline strength.
>
> ---
>
> ## Q: Limitations regarding deception and potential misuse
>
> We agree this is an important consideration. We will add a dedicated **"Broader Impacts and Ethical Considerations"** section addressing the dual-use nature of our findings. While our work demonstrates how debate guides people toward truth despite biases, we acknowledge that understanding persuasion mechanisms could potentially be misused.
>
> We will clearly discuss:
> 1. **How adversarial debate systems could be exploited** if incentives are misaligned
> 2. **Risks that persona-based persuasion insights** could enable targeted manipulation (see our response to reviewer bSCe for discussion of personalization)
> 3. **Safeguards for responsible deployment**—ensuring debate systems optimize for truth-seeking rather than mere persuasiveness
>
> Our goal is developing bias-resilient oversight that promotes accurate judgment, not enhancing deceptive capabilities. Thank you for highlighting this critical ethical dimension.
>
> ---
>
> ## Q: Formatting issue at lines 277-294
>
> Thank you for noting the formatting issue. We will ensure the final version is properly formatted according to the guidelines.
>
> ---
>
> We hope our responses—the extensive new climate study with strengthened baseline, our commitment to restructure the paper with visualization and reduced appendix reliance, and expanded discussion of broader impacts and ethical considerations—fully address the reviewer's concerns. If the reviewer finds these improvements satisfactory, we would be grateful for their consideration in revisiting the score.
>
> ### References
>
> 1. Diggelmann et al., "Climate-fever: A dataset for verification of real-world climate claims", arXiv 2020.
> 2. Gabriel et al., "Misinfo reaction frames: Reasoning about readers' reactions to news headlines", ACL 2022.

---

> > ### Comment · Reviewer_feUp · 2025-08-01
> >
> > I really appreciate the authors taking time to address each comment, criticism and suggestions in the review. After reviewing concerns from other reviewers, taking into account the additional experiments and clarifications mentioned in the rebuttal, I have updated my score for the paper.

---

> > > ### Author Response · Authors · 2025-08-01
> > >
> > > We sincerely thank the reviewer for considering our rebuttal and updating their score.

---

### Official Review · Reviewer_bSCe · 2025-07-03

**Clarity:** 4
**Significance:** 3
**Originality:** 3
**Rating:** 5
**Confidence:** 4

**Summary:**

This paper explores whether AI-assisted debate, in which two AI systems argue opposing viewpoints, can help human judges more accurately evaluate controversial factual claims related to COVID-19, compared to traditional AI consultancy (single advisor). The authors conduct two studies: (1) a human-judge experiment contrasting debate versus consultancy across different belief systems (mainstream vs. skeptical), and (2) a study using personalized AI judges designed to simulate human biases. Results indicate that debate consistently improves human accuracy and confidence calibration, especially among mainstream believers, while personalized AI judges outperform both humans and non-personalized AI judges. The findings suggest debate could provide scalable, bias-resilient oversight in critical factuality assessment tasks.

**Questions:**

- While personalized judges show higher accuracy, could they also reinforce problematic biases, echo chambers, or manipulation risks? How can we ensure personalized oversight enhances rather than diminishes epistemic integrity?
- Given debate’s strong performance, how might it integrate into practical workflows (e.g., news fact-checking)? What infrastructure or training would human evaluators require to use AI debate effectively?

**Ethical Concerns:**

["NO or VERY MINOR ethics concerns only"]

**Limitations:**

Yes

**Quality:**

4

**Strengths And Weaknesses:**

### Strengths

**\[S1] Practical significance:**
Given growing concerns around misinformation, polarization, and LLM safety, this work addresses timely and urgent issues, providing valuable evidence that structured AI debate can meaningfully improve human judgment in controversial domains.

**\[S2] Careful, transparent experimental design:**
The use of real-world, contentious claims (COVID-19), pre-screened participants with diverse beliefs, and methodical protocols (debate vs consultancy) is strong. The statistical reporting is detailed, and key confounds (e.g., prior confidence, judge experience, belief strength) are explicitly measured and modeled.

**\[S3] Nuanced, actionable findings:**
The paper does not claim universal superiority of debate, but rather documents specific scenarios where debate outperforms (especially for mainstream judges or judges with low confidence) and highlights the limitations of consultancy, especially when the consultant reinforces pre-existing (sometimes incorrect) beliefs.
Additionally, the detailed analysis of persuasion strategies, calibration curves, and confidence shifts adds considerable nuance, helping explain mechanisms behind observed improvements.


### Weaknesses

**\[W1] Limited Comparison and Benchmarking:**
The consultancy condition is notably weak (randomly correct 50% of the time), potentially overstating debate advantages. Stronger baselines, such as single-advisor conditions optimized for user persuasion or calibrated trust could substantially challenge the debate method’s dominance.
Moreover, the choice of COVID-19 claims, though justified by the authors, remains narrow. It is unclear how findings generalize to less scientific, more nuanced topics (e.g., politics, ethics).

**\[W2] Persona-based AI judges:**
While personalized AI judges provide interesting findings, limited exploration of ethical implications (e.g., personalization exacerbating biases rather than mitigating them) weakens the practical recommendation for deployment.

---

> ### Author Rebuttal · Authors · 2025-07-30
>
> We thank the reviewer for their thorough and constructive feedback. We appreciate that the reviewer recognizes our work's **practical significance** in addressing misinformation and LLM safety, acknowledges our **careful experimental design** with real-world COVID-19 claims and diverse belief groups, and **values our nuanced findings** showing specific scenarios where debate outperforms consultancy.
>
> We address the reviewer's thoughtful concerns about baseline comparisons, generalization, and ethical implications of personalized AI judges below.
>
> ---
>
> ## Q: Limited generalization beyond COVID-19 and weak consultancy baseline
>
> Thank you for raising these important concerns about generalization beyond COVID-19 and the strength of our consultancy baseline. We conducted an extensive additional study with human judges on climate change claims to directly address both issues.
>
> ### **New Climate Change Study with Stronger Baseline**
>
> To demonstrate generalization, we recruited **146 participants** (73 per belief group) from an initial pool of 1,000 based on their climate change beliefs (human-caused vs. natural cycles), yielding **421 debate sessions** and **424 consultancy sessions** after quality filtering. We selected claims from the Climate-Fever dataset [1,2] using GPT-4o screening followed by human curation to ensure they satisfy our scalable oversight criteria (details in Section 2.1.1 and Appendix B.1).
>
> Critically, we **strengthened the consultancy baseline** by allowing consultants to choose which position to argue rather than random 50/50 assignments. This addresses the reviewer's concern that our original baseline was artificially weak. Now consultants argue for the correct answer ~92.54% of the time (giving consultants significant advantage), creating a much more challenging comparison that better reflects real-world scenarios where AI assistants like ChatGPT naturally provide their best assessment rather than randomly arguing for correct or incorrect positions.
>
> ### **Results**
>
> **Table: Human judge accuracy before and after debate versus consultancy interventions for climate claims**
>
> *Belief Groups:*
> - **Belief Group 1**: Climate change is primarily caused by natural cycles and variations
> - **Belief Group 2**: Climate change is primarily caused by human activities
>
> | Group | Debate Before → After | Consultancy Before → After | Debate Advantage |
> |-------|----------------------|----------------------------|------------------|
> | **Overall** | 64.5% → **72.1%** (+7.6pp) | 65.8% → **68.3%** (+2.5pp) | **+3.8pp** |
> | **Belief Group 1** (Natural) | 51.2% → **63.9%** (+12.7pp) | 53.0% → **59.2%** (+6.2pp) | **+4.7pp** |
> | **Belief Group 2** (Human-caused) | 76.8% → **80.2%** (+3.4pp) | 74.2% → **75.6%** (+1.4pp) | **+6.0pp** |
>
> Debate achieves **72.1% accuracy** compared to stronger consultancy's **68.3% accuracy**—a **3.8% improvement**. This shows debate's advantages generalize beyond COVID-19 to other contentious scientific domains even with a substantially stronger consultancy baseline.
>
> Additionally, we conducted LLM judge studies on climate claims (detailed in Appendix D.2) which show similar patterns—debate provides better performance than consultancy across both COVID-19 and climate domains, with debate particularly benefiting weaker judges like Qwen-2.5-8B.
>
> We will update Figure 1 to include these climate results and revise Section 2.1.2 to clearly describe our stronger consultancy protocol, directly addressing the reviewer's concerns about both generalization and baseline strength.
>
> ---
>
> ## Q: Why scientific claims rather than political/ethical topics
>
> We appreciate the reviewer's suggestion about political/ethical topics. We selected scientific domains (COVID-19 and climate change) because they provide controversial claims with verifiable ground truth from scientific evidence—essential for measuring whether protocols guide judges toward factual accuracy. Political topics often lack objective ground truth and raise ethical concerns about changing people's beliefs. Our approach ensures we can rigorously evaluate protocol effectiveness while maintaining ethical standards.
>
> ---
>
> ## Q: Ethical implications of personalized AI judges
>
> We thank the reviewer for this insightful question. We agree that personalized LLM judges pose ethical risks and warrant careful consideration during real-world deployment [5,6]. While we do not endorse any collection of private user information without consent, it is well-known that social media platforms already track user behavior for personalization of feeds [7] and ad-targeting [8]. We envision that users would be able to opt-in to use of personalized debaters or judges that rely on similar tracking or user-volunteered demographic information. While LLM judges can be susceptible to manipulation risks and bias amplification, we find that the adversarial debate ensures that judges are actively exposed to arguments and counterevidence from both sides thus mitigating confirmation bias and echo chamber effects [3,4].
>
> We analyzed the LLM judge's transcripts with LLM-as-judge (Gemini-2.5-flash) to identify biased arguments (evidence of confirmation bias or echo chamber effect) and validated findings through manual checking of randomly selected transcripts, finding no significant differences to the non-personalised LLM judge. We find that the personalized LLM judges consider both the debaters' arguments when reasoning for the final verdict with no bias towards the debater that aligns with their belief personas. We will add this analysis in the final manuscript for further clarity.
>
> We will discuss the ethical considerations of personalized oversight in Section 3.2 and Section A. Specifically, we report how the personas are collected, the prompts used for the LLM judges and release the aggregate profiles for accountability and transparency. Furthermore, we leave it as our future work to develop pluralistic and multi-persona aggregation techniques on our methodology to make personalized oversight more robust.
>
> ---
>
> ## Q: Practical implementation for real-world workflows
>
> Thank you for raising this important concern about practical deployment. Based on our implementation, we can outline concrete integration pathways:
>
> ### Integration examples
> AI debate could replace single-consultant summaries in existing interfaces, such as fact-check panels on news sites or research assistants in enterprise search. Users would see a compact debate widget with 2-3 rounds of opposing arguments drawing from the same trusted sources, providing evidence and counter-evidence at a glance.
>
> ### Infrastructure and training requirements
> Minimal overhead beyond current model-serving stacks—just two parallel LLM calls per round, lightweight orchestration, and a simple UI to present arguments side-by-side. Implementation requires only API access (for closed models) or open-source model deployment with efficient frameworks like vLLM/SGLang (7B models need ~21GB memory, or 4GB with 4-bit quantization). **Human evaluators need minimal training**—our non-expert judges achieved 70.1% accuracy (vs 60% baseline) with text based bullet-point instructions. The debate format naturally exposes weak arguments since contradictions become obvious when AI agents argue opposing sides.
>
> We will **open-source our complete debate/consultancy transcripts** with both human and LLM judges to enable training specialized truth-seeking and fact-checking models. Our Discussion section will be expanded to detail these implementation pathways.
>
> ---
>
> We hope our responses—the extensive new human judge study on climate claims with stronger baseline, ethical analysis of personalized judges, and practical implementation pathways—fully address the reviewer's concerns. We would be grateful for their consideration in revisiting the score.
>
> ### References
>
> 1. Diggelmann et al., "Climate-fever: A dataset for verification of real-world climate claims", arXiv 2020.
> 2. Gabriel et al., "Misinfo reaction frames: Reasoning about readers' reactions to news headlines", ACL 2022.
> 3. Shi et al., "Argumentative Experience: Reducing Confirmation Bias on Controversial Issues through LLM-Generated Multi-Persona Debates", arXiv 2025.
> 4. Orbach et al., "Out of the Echo Chamber: Detecting Countering Debate Speeches", ACL 2020.
> 5. Dong et al. "Can LLM be a Personalized Judge?", Findings of EMNLP 2024.
> 6. Liu et al. "Evaluating Large Language Model Biases in Persona-Steered Generation", Findings of ACL 2024.
> 7. Michiels et al. "What Are Filter Bubbles Really? A Review of the Conceptual and Empirical Work," UMAP Adjunct 2022.
> 8. Zeng et al, "What factors affect targeting and bids in online advertising? a field measurement study," IMC 2022.

---

> > ### Comment · Reviewer_bSCe · 2025-08-03
> >
> > Thank you for the thorough response. Additional experiments and clarifications address the queries I had, and make me more confident in my high evaluation of this paper.

---

> > > ### Author Response · Authors · 2025-08-03
> > >
> > > Thank you for recognizing our additional experiments and responses, and for your constructive feedback that strengthened our work.

---

### Note · Authors · 2025-08-14

Dear AC & reviewers,

Thank you for the feedback and discussion. During the response period, we've addressed the majority of reviewer concerns by conducting additional experiments on climate change with human judges, confirming generalization beyond COVID-19 even with a stronger consultancy baseline. We appreciate Reviewer `bSCe` noting our additions “**make me more confident in my high evaluation**" and Reviewer `feUp` acknowledging we addressed "**each comment, criticism, and suggestion.**"

**On Reviewer bvi9's concern:** Reviewer bvi9 argues mainstream claims have more supporting evidence. This reflects real-world factuality where truth typically has more evidence. However, LLMs can be sycophantic and persuasive when arguing falsehoods [1,2], making non-mainstream believers vulnerable to manipulation.

Our study addresses this vulnerability: debate outperforms consultancy at preventing persuasion toward incorrect positions for both belief groups. While perfect evidence parity is challenging and often not guaranteed, we balanced conditions by providing equal reference sources, with credible-appearing sources supporting incorrect claims.
The suggested edge cases (unresolved claims, domains with no background knowledge) would require indefinite waiting for ground truth or remove the asymmetry central to scalable oversight. We will note these as future theoretical explorations. We chose COVID-19 and climate change because they meet all scalable oversight criteria (line 91), represent typical real-world factuality where mainstream aligns with truth, and are socially consequential domains where misinformation causes real harm.

**On Reviewer W11V's concern:** While early scalable oversight papers were preprints, the field has matured with high impact peer-reviewed work: Khan et al. (ICML 2024 Best Paper)[3], Kenton et al. (NeurIPS 2024)[4], Costello et al. (Science 2024)[5]. Our empirical findings are self-contained—extensive experiments across domains with rigorous analysis stand on their own merit.

Thank you for your consideration.

1. Sharma et al. Towards Understanding Sycophancy in Language Models, ICLR 2024
2. Salvi et al. On the Conversational Persuasiveness of LLMs, arXiv 2024
3. Khan et al. Debating with More Persuasive LLMs Leads to More Truthful Answers, ICML 2024
4. Kenton et al. On scalable oversight with weak LLMs judging strong LLMs, NeurIPS 2024
5. Costello et al. Durably reducing conspiracy beliefs through dialogues with AI, Science 2024

---

### Decision · Program_Chairs · 2025-09-17

**Decision:**

Accept (poster)

**Comment:**

This paper studied if AI debate can lead biased people to the truth, by having two AI systems debate opposing sides of controversial COVID-19 claims, topics on which people often have strong prior beliefs. Their first study used human participants with either mainstream or skeptical beliefs. They evaluated the claims using either an AI-assisted debate or a single-advisor consultancy. In their second study, the authors used personalised AI judges that mimicked the same human belief systems. In the study with human participants, they found that the debate protocol where two AI advisors presented opposing, evidence-based arguments consistently improved judgment accuracy and confidence. It also outperformed the single-advisor consultancy by 10% overall, demonstrating the effectiveness of AI debate.

Without going too much into details, this is a somewhat controversial paper, generating mixed reviews. Reviewers `bSCe` and `feUp` have a high evaluation of the paper, whereas Reviewers `bvi9` and `W11V` raised critical concerns about the paper, especially in the experimental setup. They can be summarised as follows:

- Reviewer `bvi9` argued that, "*in the chosen domain (COVID-19 claims), the true claims correspond exactly to the mainstream claims -- this makes "being mainstream" a confounding factor mediating truth and persuasion, i.e. the results can be explained simply by the mainstream view having more arguments backing it in the literature.*"

- Reviewer `W11V` pointed out that this work builds on some of the non-peer reviewed foundational work.

These two concerns are indeed hard to resolve, but overall I feel that the authors did a very good job responding to them. Regarding the first concern, the authors explained that they selected claims with verifiable scientific evidence to test if debate can help people form accurate judgments on controversial topics, regardless of their existing beliefs. They defined "mainstream" views as those aligned with scientific consensus and "skeptical" views as those that diverge from it. To further demonstrate the effectiveness of their debate protocol, they conducted an additional study on climate change claims. They also addressed a reviewer's suggestion about using prediction markets, acknowledging that such longitudinal studies could offer insights into separating truth-seeking from mainstream bias, and will include this in their future work.

Regarding the second point, the authors have already provided a clear explanation. In new and emerging fields, it's common and often necessary to build upon preliminary research, as also supported by Reviewer `feUp`.

I would therefore like to recommend this paper for publication at NeurIPS as a poster.